# Ensuring Uninterrupted MTC Service Availability during Emergencies Using LTE/5G Public Mobile Land Networks

**Padraic McKeever [1,*] , Abhinav Sadu [1], Shubham Rohilla [2], Zain Mehdi [2] and Antonello Monti [1]**

[1] Institute for Automation of Complex Power Systems, RWTH Aachen University, 52074 Aachen, Germany; asadu@eonerc.rwth-aachen.de (A.S.); amonti@eonerc.rwth-aachen.de (A.M.)

[2] Ericsson GmbH, 52134 Herzogenrath, Germany; shubham.rohilla@ericsson.com (S.R.); zain.mehdi@ericsson.com (Z.M.)

\* Correspondence: pmckeever@eonerc.rwth-aachen.de

**Abstract:** During emergencies LTE/5G-based public mobile land networks (PLMNs) restrict network access by normal users, which means a lack of service reliability which limits application of LTE/5G for machine-type communication (MTC) in critical applications, such as power systems This paper shows how existing LTE/5G features can be used to differentiate MTC of devices in a microgrid from other MTC or human-to-human (H2H) communication and ensure that these microgrid devices have service during emergencies, which enables use of LTE/5G communication to co-ordinate the use of distributed energy resources (DER) in microgrids, so that they can autarkically perform blackout recovery of an islanded microgrid. It is shown that this method allows the blackout recovery 100 times faster than with a conventional black start. The microgrid blackout recovery is demonstrated using the LTE/5G PLMN Access Barring feature. The disadvantage of using PLMN-based Access Barring is the need to define two separate PLMNs in one radio cell, which is an inefficient use of radio spectrum. However, this can be avoided by using the Extended Access Class Barring (EAB) override or application-specific congestion control (ACDC) features of the CAT-M1 low-power wide-area MTC technology, which are included in LTE and 5G standards.

**Keywords:** 5G; machine-type communication; microgrids; black start; emergency

## 1. Introduction

The operation of power systems relies on communications networks, which themselves rely on the power supply, so that power systems and their communications networks form a mutually dependent cyber-physical system. However, loss of communication can lead to power failures and power failures can lead to loss of communications, with consequential cascading failures [1]. Disasters can cause severe damage to the communications networks and also cause overload situations due to the high demand for communications during emergencies. Hybrid redundant communications systems [2] can fail if the redundant systems both depend on the grid power supply.

The aim of this paper is to show how 5G/LTE-based PLMNs can provide reliable communications in a microgrid during blackouts. Existing works on reliability of communications in power grids and microgrids do not address communication systems that continue to work with a widespread loss of mains power but assume that mains power is present [2–5].

In the context of a microgrid, having reliable communications during a blackout means that the microgrid can continue to manage itself during blackout, i.e., even when the distributed energy resources (DER) in the microgrid are connected to a de-energised grid, the distributed controllers at the DER sites can continue to communicate with each other. It was shown in [6] how an islanded

microgrid can recover from blackout on its own if its devices can communicate with each other. The approach of this paper is to use LTE/5G-based public land mobile networks (PLMN) to provide reliable communications for the exemplary microgrid blackout recovery use case. Being able to recover from blackout means that the failure modes of the power system and the communications system are decoupled, so that the cyber physical system (CPS) they form together is less vulnerable to catastrophic failure.

Our focus on how a particular concrete communications technology can be made reliable in the worst-case fills a gap in the existing work on communications reliability in smart grid applications in that it focuses on how to practically apply a specific technology to provide reliable communications in a specific use case.

The theoretical study of the topological conditions between power grid stability and communication failures performed in [7] treats communication failures without considering their cause, developing a communications topology to cope with link failures. A distributed architecture for control of a distribution grid is presented in [8], and the mitigation of failures of communications to devices at different levels of the control hierarchy is considered. However, neither of these works considers the case where there is a global communications failure, such as would occur in a blackout.

Communications reliability is treated in an undifferentiated way, i.e., considering the overall communications system reliability without considering reliability under different network operating conditions, but only the overall reliability requirements of different applications [3]. There is, to the authors' knowledge, no literature specifically considering communications reliability during blackout.

Equipping smart grid IoT devices with two UMTS Subscriber Identify Module (USIM) cards for different mobile networks will ensure continuity of service in case one network fails [9], but this method would not work if a power outage affected both mobile networks.

Literature treating specific communication requirements of smart grid applications, such as latency [4], implicitly assumes that the underlying basic communications system is actually working. This paper addresses the issue of actually ensuring a basic communications service during blackout.

Ultra-reliable communications (URC) in 5G wireless systems is discussed by Popovski [10] outlines the problem of multiple devices trying to share communication resources but does not propose a solution and does not consider how to provide reliable communications during blackouts. Providing URC for services with different latency requirements is considered in [11]. However, no distinction is made in URC literature between communication under normal power grid operation conditions and under blackout conditions.

The contribution of this paper is to show how LTE/5G PLMN communication can ensure reliable communications for selected MTC devices during emergency conditions, without giving these devices additional priority over other network users during normal conditions or adversely affecting the service to prioritised users during emergencies. This makes LTE/5G PLMN usable for critical applications where reliable communications are needed.

The paper demonstrates this by firstly developing the communication requirements to enable an islanded microgrid to recover from blackout (Section 2). Then we examine how LTE/5G PLMN communication can ensure reliable communications for selected MTC devices during emergency conditions, without giving these devices additional priority over other network users during normal conditions or adversely affecting the service to prioritised users during emergencies (Section 3). Then the use of one such LTE/5G communication feature, the PLMN-based Access Barring method, to achieve reliable MTC communication during blackout is demonstrated in a lab environment (Section 4). Finally, we quantify the reduction in recovery time, using Continuous Time Markov Chain (CTMC) modelling of the recovery of the power and communications systems from blackout by conventional means compared to the autarkic microgrid-based approach (Section 5).

## 2. Communication Requirements for Autarkic Microgrid Blackout Recovery

In this section the requirements to be fulfilled by the communications system to allow an islanded microgrid to recover on its own from blackout are developed. The autarkic microgrid blackout recovery method presented in [6] assumes that the MV/LV grid is divided into a set of microgrids (Figure 1), which can be operated in island mode and are assumed to be autarkic, meaning that they have sufficient local energy generation and storage to be self-sufficient, and autonomous, meaning that they can manage themselves, although in normal operation they may be managed by distribution grid (DG) management systems external to the microgrid. Microgrid management is performed by means of a multi-agent system (MAS), the individual agents being located at the DER sites. Microgrid management is supported by communication between the agents. The MAS could support a number of different microgrid management use cases. However, in this section, we focus on the communication requirements for autarkic microgrid blackout recovery.

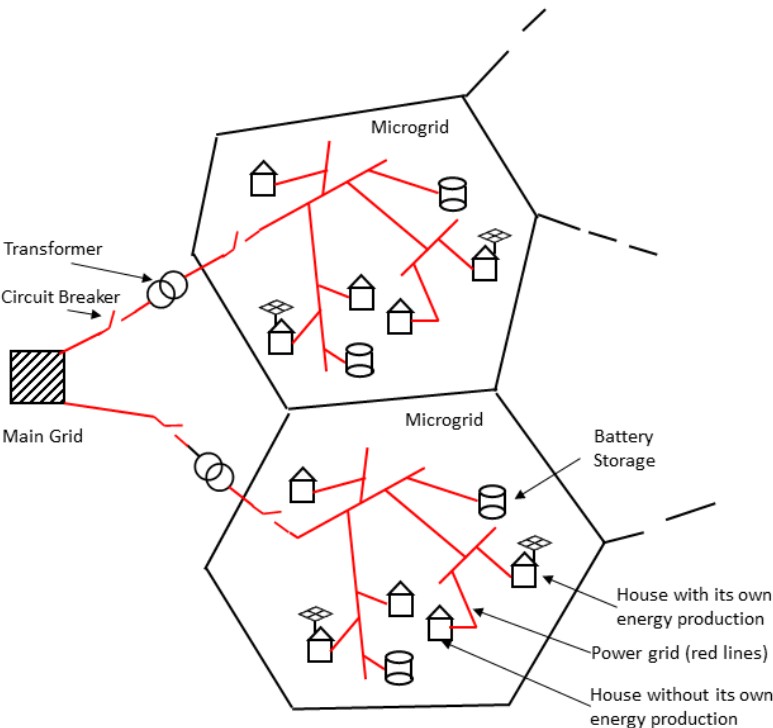

**Figure 1.** Autarkic Microgrids (each islanded microgrid can autonomously recover from blackout using only its own DERs).

If communications from the microgrid to external management systems are working, the microgrid can perform autarkic blackout recovery under their direction. If, however, there is a loss of communication from the microgrid to external management systems, the microgrid can autonomously perform the autarkic blackout recovery. In any case, the autarkic blackout recovery relies on having communications which continue to function locally in the microgrid during blackout to allow the agents to act together to perform a black start using local DERs. Hence, the main communications requirement is reliability, defined as the ability of a system or component to perform its required functions under stated conditions for a specified period of time [12]. In our black start use case, reliability of the local communication between the microgrid agents during a blackout is required. It is not required that external communication from the microgrid is working.

Availability is defined as the proportion of time that a system delivers its service. Availability is a prerequisite for reliability. Availability refers to the overall percentage of time that a system delivers its service, not its availability during blackout. While there is no specific requirement on system availability from the autarkic microgrid blackout recovery use case, better system availability under

blackout conditions is desirable. A way of improving the communications service availability for MAS MTC devices using PLMN is shown in Section 3.2 and modelled in Section 5.

Because the number of agents is large and geographically extensive, a large-scale, wide-area communications network is needed. The agents communicate using MTC [13] so the network must support MTC. Latency of 100 ms and throughput of <1 kBps per DER are acceptable [6] for the black start use case. The autarkic microgrid black start use case illustrates how reliability of the communications system during blackout can be used to make it possible that the power system can recover from failure.

## 3. Using LTE/5G-based Public Mobile Land Networks for Autarkic Microgrid Blackout Recovery

In this section, we examine how an LTE/5G-based PLMN can fulfil the requirements stated in Section 2 on communications reliability, latency and throughput for the autarkic microgrid black start use case. PLMN addresses limitations of the current communications technologies used for DG management: Internet communications generally require power to be available at the DER sites, so are vulnerable to blackouts. Using wireless technologies such as LoRa requires that the wireless network be set up to be reliable and available during blackouts. The communication range of broadband PLC is limited by its sensitivity to noise, making it unsuitable for this application.

Mobile networks fulfil the requirement for having wide-area network, while using PLMN avoids the cost of having to roll out a separate network for grid management purposes. Crucially, they also offer a high degree of reliability [3]. However, using a public network means that microgrid management traffic shares the same public mobile network with H2H traffic and MTC traffic from other applications, becoming just another application to be supported by the network, with a particular QoS requirement.

### 3.1. CAT-M1 for LTE/5G Public Network Support of DG Management Communication

In LTE [14], MTC end-user equipment (UE) contains the same USIM card as H2H traffic, so that LTE does not handle MTC traffic differently from H2H traffic in the radio network. Massive MTC applications (e.g., smart metering or sensor applications) typically send small data volumes, are latency-tolerant, but require low cost, long battery life and superior coverage in hard-to-reach areas such as basements or deep indoor environments. Support for massive MTC applications was introduced in 3GPP Release 13 [15] in the form of two LTE-based technologies: NB-IoT and LTE CAT-M1, which are suited for different types of use cases.

Narrowband-Internet of Things (NB-IoT) [15] is a low-power wide-area (LPWA) technology which provides greatly improved coverage compared to LTE CAT-M1, supports a massive number of low throughput devices, enables ultra-low device cost, and has low device power consumption. However, NB-IoT's latency can be up to 1–10 s, which is not suitable for the autarkic microgrid black start use case.

LTE CAT-M1 is a LPWA air interface that connects MTC devices with medium data rate requirements (375 kb/s upload and download speeds in half-duplex mode) and latencies of 50–100 ms, which satisfy the requirements of the autarkic microgrid black start use case. It enables long battery lifecycles and greater in-building range. It supports the EAB Override and ACDC mechanisms discussed in Section 3.4 below. As well as being in LTE, it will continue to be supported in the 5G specifications [16].

### 3.2. eNodeB Support for Microgrid Blackout Recovery

The eNodeB is the LTE/5G node which operates the radio cells. The UEs connect to the eNodeB. Blackout conditions can reduce the availability of the mobile core network. Loss of the mobile core network normally means that the eNodeBs cease to handle traffic. However, the Isolated E-UTRAN Operation for Public Safety (IOPS) [17] feature of 3GPP Release 13 [15] allows eNodeBs to continue to handle traffic within their own radio cell even if the mobile core network is unavailable. IOPS requires

additional hardware and software in the eNodeBs. IOPS enables the MAS agents to communicate as long as the eNodeB is available (regardless of the availability of the rest of the PLMN), thus enhancing the communications service availability during blackouts.

In order to give MAS MTC devices preferential handling during blackouts, the eNodeBs should know that a blackout has occurred and distinguish it from other emergency situations and congestion situations. If there is a large-scale blackout, an administrative procedure can be performed centrally in the PLMN so that all eNodeBs pre-emptively give priority access to MAS MTC devices, even if there is no local blackout. To cover the eventuality that local eNodeBs may, due to a sudden blackout, lose contact with the mobile core network and must act autonomously, eNodeBs should themselves be able to detect a local blackout. To aid their situational awareness, the eNodeBs may themselves act as agents in the MAS. Such an agent is like a UE located at the eNodeB: it has an interface towards the equipment it represents in the MAS (the rest of the eNodeB) and a communications interface towards the rest of the MAS (acting like a UE). During blackouts, eNodeBs may fail, in which case the UEs they were hosting will roam into surviving eNodeBs' cells. Network dimensioning should ensure that enough eNodeBs have standby power supplies or battery packs to support the traffic during blackouts.

*3.3. Access Barring Mechanisms*

In 3GPP mobile communications systems up to and including LTE, MTC and H2H communication compete for the same radio resources in case spectrum is shared. UEs used by normal subscribers have a randomly allocated access class (AC) value between 0 and 9 [14], which is stored in the USIM card. Specific high priority UEs may also be allocated AC 11 to 15, e.g., for emergency services.

In network congestion and emergency situations, PLMN restricts traffic by using the Access Class Barring (ACB) and/or Extended Access Class Barring (EAB), functions to bar UEs from connecting to the network. AC barring mechanisms take effect during the random access procedure (RAP) [15], which results in the UE being given an initial uplink bandwidth assignment which it uses for the radio resource control (RRC) procedure. In RRC, admission control functionality checks the UEs' QoS parameters such as QCI (QoS class identifier) and allocation and retention priority (ARP). Successful RRC results in allocation of a radio bearer and an evolved packet system (EPS) bearer, a tunnel between UE and packet gateway (PGW). Hence, to actually achieve the QoS to which it is nominally entitled, the UE must first successfully complete the RAP, which is affected by access barring in congestion or emergencies.

In ACB [14], UEs with normal AC values generate a random number which must be below a level (the Barring Factor) broadcasted by the eNodeBs for the UE to connect to the network, otherwise the UE must wait for a period defined by the eNodeB. In EAB [15], the eNodeB broadcasts an order to UEs without the required class level to stop transmitting for a period. UEs may be configured to override EAB, effectively giving them a priority over other UEs. ACB or EAB restrictions mean that MTC-type devices which are not already connected would not be able to transmit, limiting the usage of MTC to non-critical or always-connected applications.

The fact that the number of microgrid MAS MTC devices is of the same order as there are houses or people in the area precludes allocating the devices any special priority handling compared to other MTC devices or H2H users, except during the blackout. Also, giving the MAS MTC devices service during blackout should not interfere with the service to users with higher AC values, such as emergency services and so should keep AC values in the range 0 to 9.

However, during blackouts, microgrid MTC devices should reliably be able to access the network, while other normal human and MTC-type users are excluded. To achieve this, it must be possible, at least during blackouts, to bypass the network access restriction of MAS MTC devices based on their access class.

### 3.4. Preferential, Differentiated MTC using CAT-M1 in LTE and 5G

The requirement for reliability of the local MAS communications during a blackout stated in Section 2 means that the mobile access network of the PLMN must be able to distinguish the MAS MTC devices from other MTC and H2H users and give them preferential handling during network emergencies. Preferential, differentiated MTC is supported in the 3GPP standardisation since 3GPP Release 8 through the EAB Override [14] and application-specific congestion control for data communication (ACDC) [15] functions. However, these functions are currently not implemented in commercially available PLMNs and UEs. EAB Override allows UEs to access the network under EAB conditions by requesting a packet data network (PDN) connection for which EAB does not apply [15]. To ensure that MAS MTC devices can communicate during EAB conditions, MAS MTC devices could be configured with multiple PDNs, one PDN for general communication and connectivity, and second PDN specific to the blackout scenario.

ACDC applies for UEs with AC < 10 and allows or prevents connection attempts from UEs on a per-application basis. The MAS MTC devices could be defined as an ACDC application whose devices are always allowed to connect, fulfilling the microgrid black start communication reliability requirement.

Hence, either of the CAT-M1 EAB Override and ACDC functions could ensure that MAS MTC devices can communicate reliably during emergencies. ACDC has the advantage over EAB Override of not requiring multiple PDNs. Both ACDC and EAB allow reliable MTC communication during blackout to be achieved while using radio resources more efficiently than the PLMN-based Access Barring described in the next Section 3.5, as they allow the microgrid MTC devices to use the same PLMN as other MTC and H2H devices. In the long-term, MAS use cases, including autarkic microgrid black start, can be run on CAT-M1 (massive IoT) networks, which are part of both the LTE and 5G standards.

### 3.5. Preferential, Differentiated MTC Using PLMN-Based Access Barring in LTE

In LTE, the UE is responsible for selecting a PLMN for subsequent cell selection. A PLMN consists of a radio access network and a core network, and has a unique id. A UE can only be in a single PLMN at any time. A single cell can belong to multiple PLMNs, so the network may broadcast a list of PLMN identities. Separate access control parameters can be set for each PLMN id for network sharing.

In this method, the radio spectrum resources are divided between two separate PLMNs with separate PLMN ids. We use network sharing access control (based on PLMN id), which allows access barring to differentiate based on PLMNs in a mobile network cell. One PLMN supports the MAS MTC devices, which shall receive preferential access barring treatment and continue to be able to connect to the network and communicate in the particular emergency situation of a blackout. The second PLMN supports normal smartphone users and other MTC devices which shall be subjected to ACB restrictions during the blackout.

When there is a blackout in the islanded microgrid, the eNodeB will apply ACB restrictions in the second PLMN. The first PLMN (with the MAS MTC devices) will not be subject to ACB restrictions during the blackout. In this way, the method results in the required preferential, differentiated handling of the MAS MTC devices during emergencies. This method allows the operator to easily, quickly and dynamically apply preferential policies for the MAS MTC devices, differentiating them from the normal UEs and other MTC devices in the network. This method is functionally equivalent to using EAB Override or ACDC but has the disadvantage that it means statically pre-allocating scarce radio channel resources, which means that this solution is not generally viable. Also, assigning different PLMN ids in the network has the disadvantage that it is not a scalable solution when different type of MTC devices and services are supported by the network: the number of different PLMN ids is limited to roughly 50 per country, so that the number of different ways of handling different MTC use cases is also limited. An advantage of the method is that ACB and multiple PLMNs are supported in currently commercially available LTE networks and device ecosystems.

## 4. Laboratory Test of PLMN-Based Access Barring for Communications Reliability during Blackout Recovery

The test system setup is shown in Figure 2. The lab tests in this work mimic the situation where an LTE network provided by a telecom operator is shared between an Energy Utility's MTC network and other users (including other MTC devices, normal Smartphone users and users with ACB > 10). The tests were performed using a commercially available LTE system, set up to make a private mobile network in a lab environment.

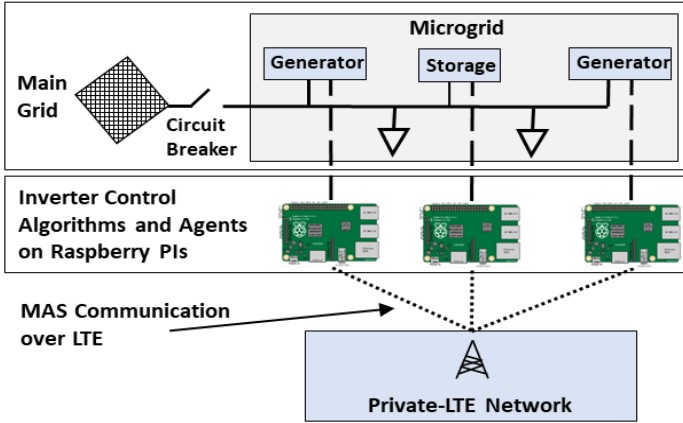

**Figure 2.** Test setup for PLMN-based Access Barring lab test.

The Private LTE system consisted of Evolved Packed Core (EPC), which provided access to external packet IP networks and perform several core network related functions (e.g., QoS, security, mobility and terminal context management); User Data Consolidation (UDC), which provided management of subscription data; Radio and Transport components; Enterprise Operations Support System (E-OSS) which provided local operations for enterprises; Remote Radio Units and Indoor Radio Units for connecting to the user equipment. The private LTE system established data connectivity between the user equipment and a connected external (private) mobile network. The private LTE system connected with one antenna. It provided all the functionality of LTE but with a small footprint and compact deployment. The modulation technique used was adaptive modulation and coding (AMC), so that the exact modulation changes based on the radio signal strength, direction, distance of the user.

CAT-M1 functionality was not available in the test LTE network, so that the tests were limited to the PLMN Access Barring method.

The test electrical DG grid is a real-time simulation of an AC microgrid, connected by a circuit breaker to the main grid. As tested, the microgrid consisted of four photovoltaic generators and four battery storage systems feeding four dynamic loads. The generators and storages are connected to the grid via inverters which implement a distributed consensus-based secondary control algorithm [6]. The inverter control algorithms are implemented on Raspberry PI microcomputers which receive grid measurements from the grid simulation and send control signals back. Each inverter interfaces to an agent on the same Raspberry PI. The agents form a MAS, implemented using the Calvin framework [18]. The Raspberry PIs have dongles through which the agents communicate peer-to-peer with each other over a private LTE network to implement the autarkic microgrid black start use case.

Two PLMNs are configured in a single network cell, the first PLMN serving the MAS MTC devices (instantiated in dongles inserted in the Raspberry PIs hosting the individual agents) and the second PLMN serving a number of normal smartphones. The MAS MTC devices are attached on PLMN1 and the smartphones on PLMN2. Then a simulated microgrid blackout was initiated. In the test, we assume that this blackout leads to emergency conditions in PLMN2, leading to blocking of normal subscribers by setting the Barring Factor to 0 for ACs 0 to 9. In PLMN1, no ACB restrictions were set for the MAS MTC devices during the blackout emergency.

The observed result was that only PLMN1 UEs i.e. MAS MTC devices could access the cell, and that PLMN2 UEs were barred from accessing the network. The tests confirmed that, by having a shared LTE network environment with two separate PLMNs configured in the network cell, different Access Barring treatment can be provided to different UEs registered in different PLMNs by broadcasting different Access Barring factors to the different PLMNs' devices.

The lab tests confirmed that the method enables the MAS MTC devices to receive temporary preferential treatment under particular emergency circumstances so as to be able to continue to communicate with each other during the blackout, enabling a black start to be performed in the islanded microgrid.

### 4.1. Measurement of Latency

Latency measurements between two MAS MTC devices (USB dongles connected with the Raspberry PIs in the above test setup) were performed by pinging from one device to the other and capturing and logging the traffic on both devices with tcpdump. The packet size was 56 bytes plus 8 byte of header = 64 bytes to be sent between the UEs via the mobile network nodes shown in Figure 3, so that the latency includes that of the private radio access network and private mobile core network outside the lab's radio access network. There was no other traffic (public users' traffic) on the network during the test, which reduces the latency to the minimum possible. Each test was done with 20,000 packets, with a periodicity of 1, 10 or 100 ms.

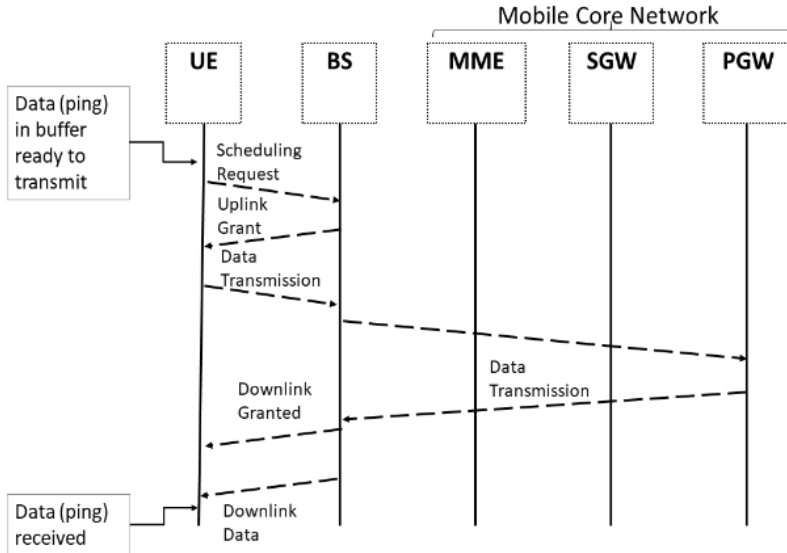

**Figure 3.** Packet Flow in Latency Tests.

Figure 4 shows that average latency achieved is 20 ms. With periodicity of 10 ms, more than 3000 packets were received within 22 ms delay. The maximum latency observed is 40 ms. This includes (Figure 3) the latencies of the two radio links, the core network and the message flow of packets over the private LTE network.

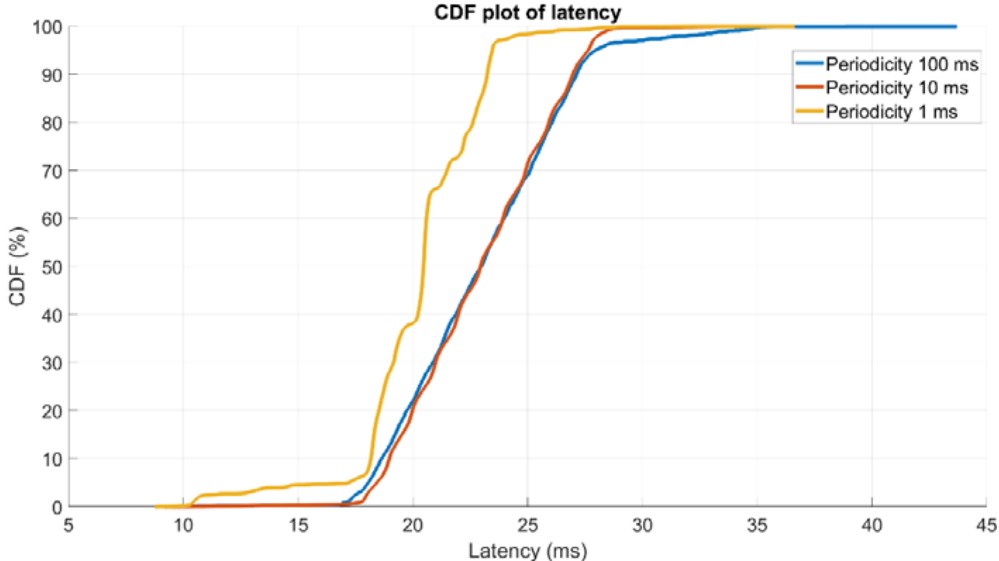

**Figure 4.** LTE Latency Measurements in Test Setup.

The result was that 95% of packets were received within 23 ms and the maximum latency was 40 ms, satisfying the use case's latency requirement.

*4.2. Evaluation*

It has been shown that PLMN specific access barring feature can bar the Smartphone devices in emergency situations and give simultaneously access of the network to smart grid MTC devices, thus providing a resilient communications solution. It has also been shown that the LTE can provide latency of below 40 ms for smart grid devices. The results demonstrate that the PLMN specific Access Barring feature is directly applicable to real LTE networks. The lab tests demonstrate reliable MAS MTC communications during emergencies by preferential, differentiated use of an LTE mobile network. The reliable communications allowed the microgrid to autonomously perform black start in an islanded microgrid, while other communications network users lost service.

## 5. CTMC Estimation of improvement in Blackout Recovery Time with Autarkic Microgrids

In this section, continuous time Markov chain (CTMC) modelling of the power grid and communications network is used to quantify the expected improvement in the blackout recovery time using the autarkic microgrid method [6] compared to the conventional top-down black start method. CTMC is a state space based stochastic modelling technique to calculate the reliability and availability of a system given the stochastic operational status of its constituent components [19]. It is specifically used for systems in which the operational state of a component depends statistically on the operational state of other components.

Grid restoration after a large-scale blackout is conventionally done by first starting power stations with black start capability, then expanding the re-energised grid to include more large synchronous power stations, and expanding the re-energised grid. In this way, MV and LV levels are re-energised hierarchically and top-down, i.e., from the transmission level. The CTMC model of conventional grid restoration is based on the following assumptions, as shown in Figure 5a:

- the DERs in the DG act independently without external control and without external communications;
- in case of blackout, they disconnect from the power line;
- the substations receive control signals from, and send measurements to, a DG grid control centre;
- the areas enclosed by dotted lines correspond to the grids served by the substations;

- the substations and the control centre have a private communications network available, and this network continues to operate during blackouts.

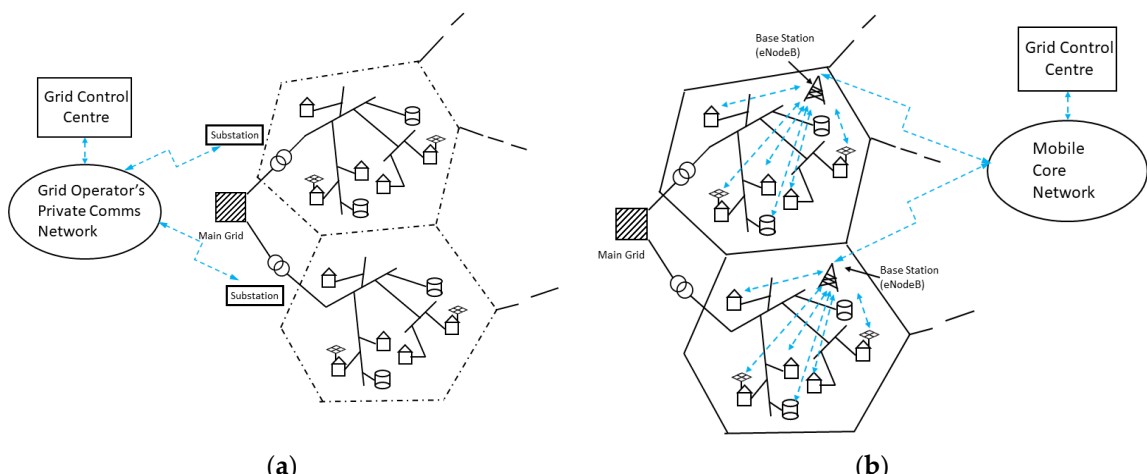

**Figure 5.** (**a**) Conventional grid restoration (hierarchical top-down blackout recovery) and (**b**) autarkic microgrid restoration (each islanded microgrid autonomously recovers from blackout).

In the autarkic microgrid blackout recovery method the MV/LV grid is divided into a set of autarkic microgrids, as shown in Figure 5b, which are islanded in case of blackout. The DERs remain connected to the grid and continue to use their MAS MTC devices to communicate with each other and with the DG control centre during the blackout [6]. To simplify the modelling, we assume each microgrid has its own eNodeB equipped with an emergency power supply. Two test cases are modelled: Test Case 1 compares the blackout recovery times of the conventional and autarkic microgrid methods; Test Case 2 compares the availability of communications in the autarkic microgrid with and without the IOPS function.

- **Test Case 1, Scenario 1: Conventional hierarchical top-down blackout recovery:** no use is made of DERs present in the MV and LV levels to assist in the blackout recovery. The simplifying assumption is made that the communications network continues to function in the blackout. The blackout recovery steps are: to re-energise the grid control centres (SCADA) and bring up generation units; to re-energise the transmission grid; and to re-energise the DG and loads.

**Table 1.** CTMC States in conventional hierarchical top-down blackout recovery (Scenario 1).

| State | State Meaning |
|:-----:|:-------------:|
| S1 | Blackout |
| S2 | SCADA_Up, Generator_Dn,Network_Dn, Load_Dn |
| S3 | SCADA_Dn, Generator_Up,Network_Dn, Load_Dn |
| S4 | SCADA_Up, Generator_Up,Network_Dn, Load_Dn |
| S5 | SCADA_Up, Generator_Up,Network_Up, Load_Dn |
| S6 | Grid Restored |

The different CTMC states and mean state transition times are tabulated in Tables 1 and 4 and shown in Figure 6. The state transition sequence and the mean time taken for individual steps of the restoration process are derived based on the recovery from the Italian blackout of 2003 [20] and the US-Canada blackout of 2003 [21].

**Table 2.** CTMC States in autarkic microgrids blackout recovery (Scenario 2).

| State | State Meaning |
|---|---|
| S1_MG | Blackout |
| S2_MG | MG controller up, real time MG capability and topology identified |
| S3_MG | Microgrids restored |
| S1_MG | Blackout |

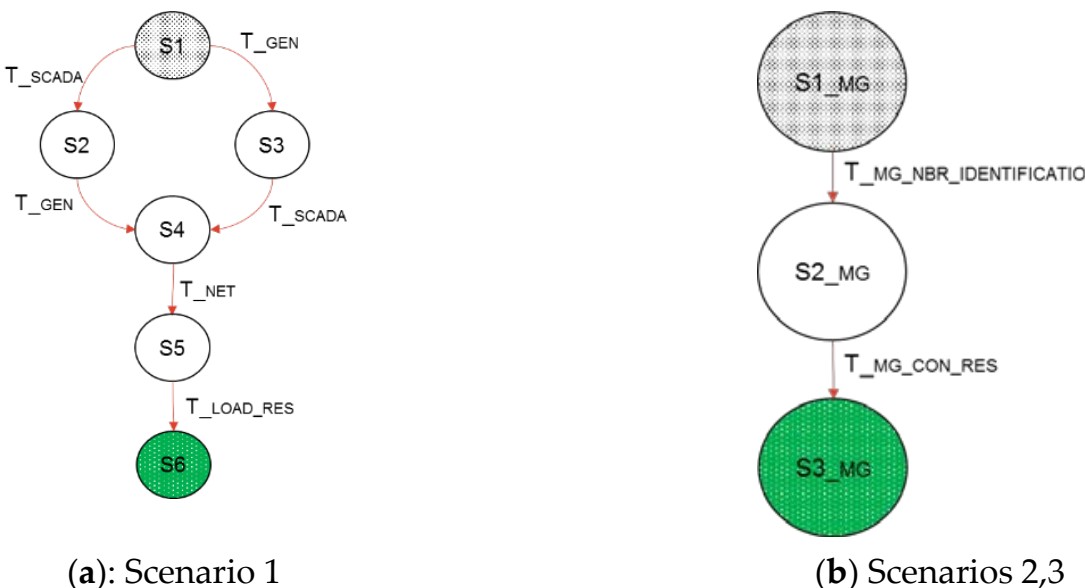

(**a**): Scenario 1        (**b**) Scenarios 2,3

**Figure 6.** (**a**,**b**) CTMC Models for Test Case 1.

**Table 3.** CTMC Transitions in Autarkic Microgrids Blackout Recovery (Scenario 3).

| Transition | Transition Purpose: Time to | Value (hours) |
|---|---|---|
| T_MG_NBR_IDENTIFICATION | bring up SCADA | 0.02 |
| T_MG_CON_RES | bring up the generators | 0.08 |

**Table 4.** CTMC Transitions in conventional hierarchical top-down blackout recovery (Scenario 1).

| Transition | Transition Purpose: Time to | Value (hours) |
|---|---|---|
| T_SCADA | bring up SCADA | 1 |
| T_GEN | bring up the generators | 10 |
| T_NET | energise the transmission grid | 4 |
| T_LOAD_RES | restore the load | 6 |

- **Test Case 1, Scenario 2: blackout recovery using autarkic microgrids:** autarkic blackout recovery is performed in the microgrid using MTC-based MAS communications but without the IOPS function, so that, if the eNodeB loses service from the mobile core network, the communication in the MAS fails. It is assumed that the eNodeBs are equipped with emergency power supplies, so that they continue to provide service in the blackout. Here, the microgrids autarkically perform black starts. The steps in the blackout recovery procedure in the microgrid are, firstly, to identify which nodes in the MAS are working; secondly, to identify the current microgrid topology and DER capacity; and, thirdly, to re-energise the microgrid and the loads. The CTMC states and conservative estimates of the state transition values are given in Tables 2 and 5.

- **Test Case 1, Scenario 3: Improved blackout recovery using autarkic microgrids and continuous preparation for blackout**: this is a further development of Scenario 2, but with the MAS continuously preparing during normal operation for blackout. It continuously monitors which agents are reachable, and deduces the current microgrid topology and what the DER production is. Based on this situational awareness, it knows whether it is able to perform black start in case of all possible combinations of nodes failing. Table 3 shows the neighbour identification time (T_MG_NBR_IDENTIFICATION), which includes the identification of the working MAS nodes and the actual grid topology. Furthermore, the value assumed for the time to coordinate the different MAS nodes to restore the microgrid (T_MG_CON_RES) is also shown.

**Table 5.** CTMC Transitions in autarkic microgrids blackout recovery (Scenario 2).

| Transition | Transition Purpose: Time to | Value (hours) |
|---|---|---|
| T_MG_NBR_IDENTIFICATION | bring up SCADA | 0.75 |
| T_MG_CON_RES | bring up the generators | 2 |

Figure 7 shows the mean blackout recovery time and the probability of blackout recovery for the three scenarios of Test Case 1. It can be seen that the grid is restored faster with the autarkic microgrid-based blackout recovery where individual microgrids are energised in parallel and quicker than in the traditional top-down method of Scenario 1. The mean restoration time is reduced by a factor of the order of 10 for the restoration with autarkic microgrids (Scenario 2) and reduced further by more than 10 times for the restoration with autarkic microgrids with continuous preparation (Scenario 3), where the steps of restoration are faster due to having pre-processed information regarding the identification of the available MAS nodes and the consequent contingency.

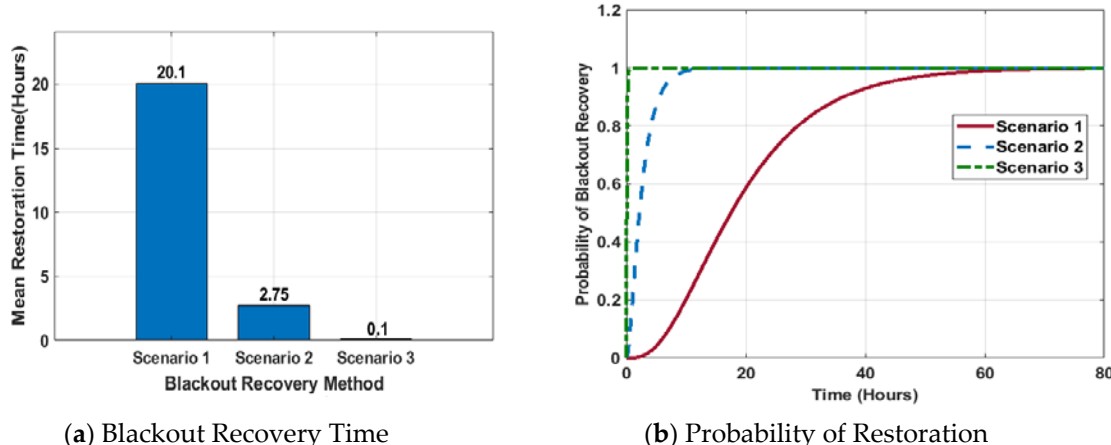

(**a**) Blackout Recovery Time      (**b**) Probability of Restoration

**Figure 7.** Test Case 1: (**a**) Blackout Recovery Time and (**b**) Probability of Restoration for Conventional and Autarkic Microgrid Scenarios.

Test Case 2 shows how the availability of the communication infrastructure is improved with the public safety (IOPS) function [17]: as shown in Figure 8, IOPS ensures that eNodeB continues to function if the mobile core network fails.

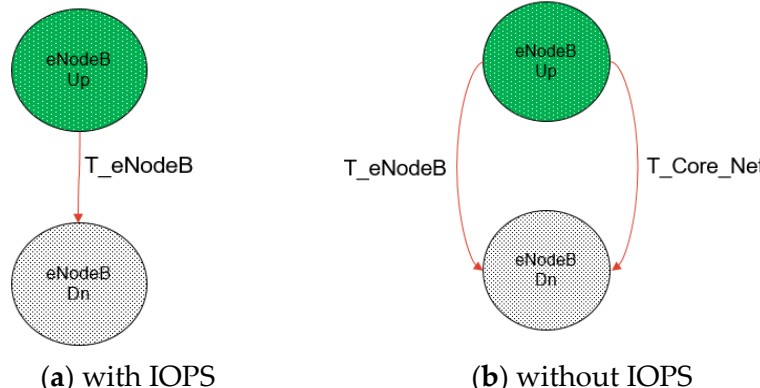

(**a**) with IOPS                          (**b**) without IOPS

**Figure 8.** Test Case 2: CTMC models of communication link failure model (**a**) with and (**b**) without IOPS.

When the eNodeB is configured with IOPS, the communication link between the microgrid devices, which traverses the eNodeB, fails only if the eNodeB fails. This is because all the core network functionalities are deployed within the eNodeB. In the case of absence of IOPS functionality, the intra-MAS communication link fails when either eNodeB fails or the core network fails. The stochastic failure model of the two eNodeB configurations is shown in Figure 8. The failure probability distributions are modelled as exponentially distributed [22].

The eNodeB failure rates assumed are given in Table 6, derived from the failure rates proposed in [23]. The probability of intra-MAS communication link failure and the mean time for the link to fail is shown in Figure 9. The results show the reduced intra-MAS link failure for the eNodeB with IOPS. The mean-time-to-failure of the eNodeB with IOPS is almost three times the mean-time-to-failure without the IOPS, increasing the intra-MAS communication link availability accordingly.

**Table 6.** Core Network, eNodeB mean times to failure [23].

| Transition | Transition Purpose: Time to | Value (hours) |
|---|---|---|
| T_eNodeB | Mean time for eNodeB failure | 0.002 |
| T_Core_Net | Time to bring up the DERs generation | 0.003 |

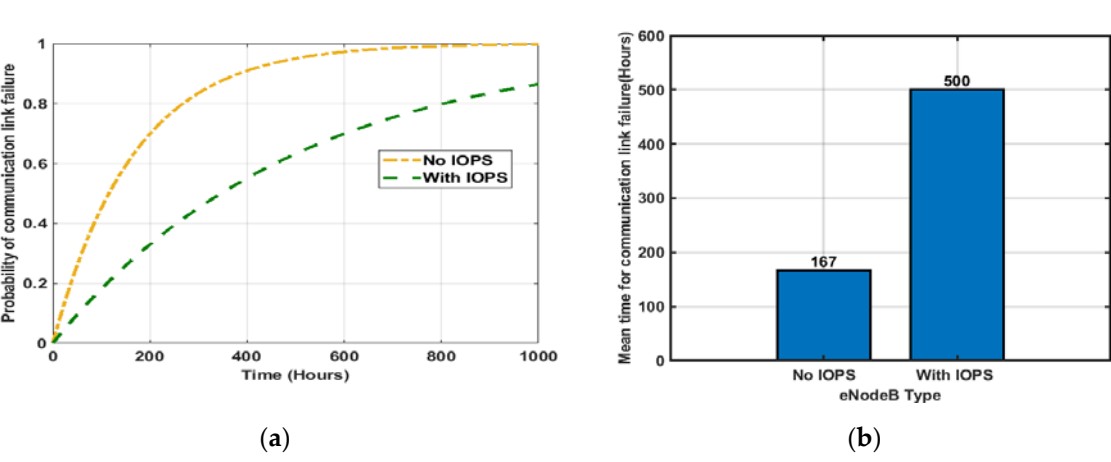

(**a**)                                          (**b**)

**Figure 9.** Test Case 2: Probability (**a**) and mean time of failure (**b**) of the MAS communications with/without IOPS.

## 6. Conclusions

Reliable communication is the enabler of black start in autarkic DER powered microgrids. LTE/5G-based PLMNs can be used to provide reliable communications to MTC devices in a microgrid

even during a blackout by differentiating the microgrid MTC devices from other MTC and H2H users and giving them preferential handling in emergencies.

Using LTE/5G-based PLMN for microgrid communications allows re-use of any existing PLMN which covers the area of the microgrid. This means that reliable MTC can be achieved without having to build a separate network, which cannot be achieved with other wireless technologies such as LoRa.

CTMC modelling of the use of PLMNs shows blackout recovery time being reduced by a factor of the order of 100 compared to the conventional black start method. The direct consequence is reduced power outage duration locally in the microgrid. Furthermore using the DERs for blackout recovery will reduce the need for central generators to provide black start ancillary service.

PLMN-based Access Barring, which is available in commercial PLMNs, can provide this preferential, differentiated handling and its use to achieve reliable MTC communication during blackout is demonstrated in this work. However, its use requires an inefficient static pre-allocation of radio network resources to the microgrid MTC devices into a separate PLMN. In future, reliable MTC communication during blackout may be achieved more efficiently by using the Enhanced Access Barring (EAB) Override or Application specific Congestion control for Data Communication (ACDC) features instead, when these features are available in commercial PLMNs.

**Author Contributions:** Conceptualization, P.M., A.S. and S.R.; methodology, P.M., A.S. and S.R.; software, A.S. and Z.M.; validation, A.S. and Z.M.; formal analysis, A.S.; investigation, P.M., A.S. and S.R.; resources, S.R.; data curation, A.S., Z.M.; writing—original draft preparation, P.M., A.S.; writing—review and editing, P.M.; visualization, P.M., A.S.; supervision, P.M.; project administration, P.M.; funding acquisition, A.M. All authors have read and agreed to the published version of the manuscript.

**Funding:** This research was funded by the German Ministry for Industry and Energy, grant number 03ET7549A).

**Conflicts of Interest:** The authors declare no conflict of interest. The funders had no role in the design of the study; in the collection, analyses, or interpretation of data; in the writing of the manuscript, or in the decision to publish the results.

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
