# Peer review of "Ensuring Uninterrupted MTC Service Availability during Emergencies Using LTE/5G Public Mobile Land Networks"

_telecom, doi:10.3390/telecom1030013_

Round 1

Reviewer 1 Report

This paper is about using LTE/5G network fo blackout microgrid emergency recovery. Although the LTE/5G is considered here for means of communications between the nodes, this paper bring very little contribution to the topic of telecommunications. The only part focused on telecommunication is the review of the available configuration options of LTE to maintain the continuity of MTC communication during blackout. The evaluation part does not bring any contribution to the field of telecommunication, with only the possibility to use LTE for microgrid recovery tested. I think the proper research field for such paper is rather power engineering than telecommunications.

Author Response

As use of renewable energy sources increases, the importance of telecommunications in power systems rises, especially in distribution grids. If renewable energy sources are widely distributed (e.g. in most houses in a city), then it will make sense to use existing communication networks, such as mobile PLMN, rather than build some dedicated network. This paper is about a use case of telecommunications for power systems. The power system aspects of this use case are dealt with in the reference [6]. The telecommunication aspect is the subject of this paper and examines how to use existing features of LTE/5G for this use case.

The reason for submitting the paper to this journal was that the journal advertises on its “About Telecom” that it covers use cases and also mobile communications 5G, 6G. So it seemed to be a suitable journal.

Reviewer 2 Report

The authors investigate an important use case for the efficient operation of smart grids, i.e. recovery during black outs. Towards this end, they provide results using LTE PLMN for machine-type communications (MTC). Overall, the topic is important and the results using an experimental test-bed are interesting. Nonetheless, there are several limitations that must be addressed by the authors in order to enhance the contribution and scope of the manuscript:

1. In the introduction, the authors should expand the discussion of relevant works in order to specifically illustrate their limitations and the gap that the authors aim to fill, as described in page 2 . Overall, the state of the art review seems limited, as more methods should be reviewed in order to provide a broad view on achieving ultra-reliable communication for smart grids. Furthermore, classification of these methods and the use of tables should be considered by the authors.

2. Without the LTE CAT-M1 functionality in the test network it is difficult to asses how the two options stack up against each other. Moreover, the discussion on the use of LTE CAT-M1 can not be evaluated without experimental data.

3. More details on the distributed consensus-based secondary control algorithm of [6] and the Calvin framework [17] must be provided in Section 4 here to facilitate the readers.

4. Why the authors set the barring factor to 0, thus blocking H2H traffic? Results for intermediate values can better depict the efficiency of the proposed technique in cases where some UEs requires connectivity. In its current form, the evaluation setup seems ideal for the purpose of black out recovery.

5. Additional details on the laboratory setup should be given, such as the adopted coding and modulation, the number of antennas etc. Also, results for different configurations of such parameters could be provided in order to show how the proposed algorithm performs.

6. Other relevant solutions should be included in Sections 4-5 in order to perform comparisons with the proposed autarkic scheme.

7. Additional metrics should be evaluated in order to illustrate the network's behaviour during the blackout recovery period? For example, throughput BER/PER and average latency would provide a more broad view on the proposed solution's performance.

Minor issues:

-Some descriptions can be simplified, e.g. at the start of Section 2 it is not necessary to refer to Section 3 since it has been already given at the end of Section 1.

-Figure 1 should be improved by including in it, descriptions of the different entities

-Several presentation issues exist, such as: bold Table 3 and regular Table 4 one the top of page 10, a non existing reference at the start of Section 4, Figure 4 appears twice etc.

Author Response

Thank you for the thorough review and the encouraging comments.

Point 1: I have expanded the treatment in lines 50-78, including a review of ultra-reliable communications in 5G. I have found no literature similar to the topic of this paper (making a communications system work reliably for MTC in a blackout). This is the reason for the limited treatment: to my knowledge, there are no equivalent works to my paper which I can survey and classify.

Papers I have seen on URC do not focus in on the needs of particular use cases but rather deal with the general case. For example,the editorial comments to an IEEE JSAC special issue of April 2019 (https://ieeexplore.ieee.org/abstract/document/8667939) acknowledge that different applications will have different requirements, but the focus is on covering the general case. I have not found any literature dealing with particular use cases and this is the gap which I point out.

Point 2: I am afraid that, as I state in lines 287-288, CAT-M1 functionality was not available in the advanced test network we used, although it had the latest available software from the Ericsson R&D organisation. This is due to a lack of demand for the functionality from network operators, so that this already standardised functionality has not yet been implemented in commercial products. The reason for their being no such demand, of course, is that the operators don’t see the use of it yet, due to a lack of use cases such as the one covered in my paper! Of course, if CAT-M1 had been available, we would have tested it and reported on it. Such a test is something for a further project and is outside the scope of this paper. The current paper tested the latest available functionality, kindly provided by the telecom equipment manufacturer Ericsson.

Point 3. In this paper, I concentrate on the communications aspects and provide references to papers dealing with other aspects of the use case. The reader can consult these referenced papers for more details of these related but extraneous matters, which do not form the subject of the current paper, whose focus is on making communications reliable during blackout. The inclusion of explanations of power systems and multi-agent system aspects would require the addition of significant text in this paper, resulting in it being harder for the reader to grasp the message of this paper, without adding any new material that has not already been published. I have therefore deliberately limited myself to providing references, where the interested reader will kind all the necessary information.

Point 4: Setting the barring factor 0 was used to test that the H2H subscribers were blocked. We performed the test in a lab without any real H2H users, and just a couple of smartphones, so this seemed adequate. A better test would be to have a traffic generator to emulate the H2H traffic, but such equipment was not available (it is considered too valuable by Ericsson). Thanks for your suggestion, I agree it would be better to test intermediate cases. I guess we could have rigged up a system to generate H2H traffic using, for example, transfer of large files by ftp, and experimented with different settings of barring factor, but such tests have not been performed.

Point 5: I have added lines 276-286. We used a portable unit called a “flight box”, which contains an eNodeB and edge cloud functionalities in our lab. The “flight box” generates the PLMNs to which the devices attach. Its antenna is located in a separate room in our lab in RWTH. The HQ for development of Ericsson’s mobile systems is physically near our lab. Ericsson have extensive labs there and we used a test mobile core network physically located in Ericsson’s premises, connected with the “flight box” located in our RWTH lab.

Point 6: I’m afraid that we have just tested this one solution, so regret that I can’t satisfy this comment. I guess that, if this paper is published, then some alternative solutions might be developed in future, especially as the issue of blackouts is quite fundamental.

Point 7: I have added Ch. 4.1 with latency test results, at line 316 et seq. I did not include such results before, as they are not really relevant to the test performed, which is testing the binary question of whether the proposed solution works at all, where we verified that it does work, rather than the quality of the communication (latency, BER, PER). However, the test was performed with a very small traffic load, and the latency is determined by the lab setup and has no real relevance beyond the lab test, i.e. no scientific value. it effectively represents an experimental measurement of the minimal latency to be expected in such a network.

Minor issues: I updated the start of Section 2 at line 95 as you suggested. I updated Figure 1 with descriptions of the different entities. Although I am happy to fix any such issues, I’m afraid I could not find some of the things you list:

  • Regarding your comment on Table3 and Table 4, their different parts are handled exactly the same, I can’t see the issue you refer to.
  • Regarding your comment on a non-existing reference at the start of Section 4: the only reference at the start of Section 4 is to Figure 2 and this reference exists, so I cannot find the mistake you refer to.
  • No Figure appears twice, also not Figure 4 on line 332 in the version sent for review. Figure 4 is similar to Figure 6, is that what you mean? But they are different Figures.

Round 2

Reviewer 1 Report

The paper has been improved with some evaluation of the proposed communication network in terms of latency. Still the paper focuses more on blackout recovery of microgrids, however, in current form it brings more research results in the area of telecommunications.

Reviewer 2 Report

The authors have addressed my concerns. I am happy with their effort.